# Enhanced Recovery after Surgery (ERAS) Protocol for Early Discharge within 12 Hours after Robotic Radical Hysterectomy

**DOI:** 10.3390/jcm11041122

**Published:** 2022-02-20

**Authors:** Jiheum Paek, Peter C. Lim

**Affiliations:** 1Division of Gynecologic Oncology, Department of Obstetrics and Gynecology, School of Medicine, Ajou University, 164 Worldcup-ro, Yeongtong-gu, Suwon 16499, Korea; paek.md@gmail.com; 2Center of Hope, Reno School of Medicine, University of Nevada, 75 Pringle Way, Reno, NV 89502, USA

**Keywords:** robotics, enhanced recovery, cervical cancer

## Abstract

To evaluate safety of quick discharge after robotic radical hysterectomy (RRH) in a tertiary hospital which has the enhanced recovery after surgery (ERAS) protocol. Among 94 consecutive cervical cancer patients who had undergone RRH, operative outcomes and the rate of unexpected visit after surgery were analyzed retrospectively. Patients were categorized as a surgery-to-discharge time of ≤12 h (early discharge [ED]) or >12 h (late discharge [LD]). About 77% (*n* = 72) of analyzed 94 patients discharged within 12 h after RRH. The ED group had significant correlation with shorter duration for urinary catheter required, less operative blood loss, and less voiding difficulty after long-term follow up compared to the LD group. There was no difference of perioperative complications and unexpected visit between the two groups. Performing nerve sparing (NS) RRH was only independent predictor for ED (*p* = 0.043, hazard ratio for LD = 0.22, confidence interval = 0.05–0.95). In conclusion, the ED within 12 h after RRH was safe in the setting of ERAS protocol. The NS-RRH could avoid the delay of genitourinary function recovery after surgery which caused LD. It can become the reasonable clinical pathway to discharge early patients who undergo NS-RRH with ERAS protocol.

## 1. Introduction

Enhanced recovery after surgery (ERAS) programs are evidence-based multimodal perioperative protocols designed to get quick recovery after surgery. The integral parts of ERAS protocols consist of preoperative counselling, optimization of nutrition, standardized pain control, and early mobilization [1]. Since an ERAS program was first introduced in 1997 [2], there have been a great number of studies which showed it reduced operative complications and facilitated rapid recovery after surgery [3,4]. In the gynecologic field, the ERAS protocol has been implemented widely for patients with gynecologic cancers as well as benign disease [5,6,7]. In addition, recommendation guidelines for perioperative care in patients who underwent gynecologic surgery have been updated [8,9]. However, there is limited data which showed the ERAS program improved clinical outcomes and quality of care in patients who underwent radical hysterectomy (RH). RH is one of the most complex procedures in surgical treatment for gynecologic disease. Patients who undergo RH have suffered voiding difficulty because of the disruption of the inferior hypogastric plexus by parametrium resection [10]. Consequently, it could slow down recovery after surgery.

Due to the outstanding development of surgical instrumentation and technique, robotic surgery has been performed widely as minimally invasive surgery (MIS) [11,12,13]. For the RH, robotic surgery has enabled early bladder function recovery and feasible outcomes after surgery [10]. Because the technical advantages of robotic surgical system, including improvement of surgical precision, visualization, and ergonomics [14], have helped surgeons perform surgical procedures in the deep pelvic cavity easily. Based on these considerations, the concept of combining ERAS program and robotic surgery seems to be an ideal approach for rapid recovery after surgery in patients who undergo RH. In addition, an objective outcome is needed to be evaluated to determine the feasibility of ERAS program. The aim of this study was to evaluate safety of quick discharge after robotic RH (RRH) in a tertiary hospital which has the ERAS protocol.

## 2. Materials and Methods

This study was approved by the Institutional Review Board at University of Nevada, Reno. Between April 2008 and July 2018, we performed RRH in 94 patients with cervical cancer. In this study, all of them were enrolled without exception and we reviewed their medical charts retrospectively. Patients were categorized as a surgery-to-discharge time of ≤12 h (early discharge [ED]) or >12 h (late discharge [LD]). The patient’s status was estimated in terms of the type of RH, operating time, estimated blood loss, perioperative complications, days of urinary catheter required, the rate of visit to emergency room after discharge, and the rate of re-admission after surgery. In addition, their subjective genitourinary symptoms were evaluated at 2 years after surgery. Patients’ pain was controlled by the pain management of our ERAS protocol (Table 1). The urinary catheters of all patients were removed immediately after surgery according to the clinical pathway in the institution [10]. If postvoid residual urine volume (PVR) measured after spontaneous voiding was less than 150 mL, they could leave the hospital. If they could not void or their PVR > 150 mL, they were discharged after recatheterization or stayed at hospital overnight to check both spontaneous voiding and PVR in the next morning. Additionally, we classified complications into minor and major complications. Minor complication included urinary tract infection with fever >38.5 °C. Major complications included the situation requiring a secondary surgical procedure to perform adequate hemostasis and repair of urinary tract injuries or bowel perforation.

All continuous data were expressed as mean ± standard deviation, and categorical data were reported as an absolute number or percentage. Frequency distributions were compared using the Chi-square test and Fisher’s exact test. The mean or median values were compared using the Student’s *t*- and Mann–Whitney U-tests. All calculated *p*-values were 2-sided, and *p* < 0.05 was considered statistically significant. Data were analyzed using the SAS/STAT software, version 9.4 (SAS Institute Inc., Cary, NC, USA).

## 3. Results

A summary of clinicopathologic characteristics was described in Table 2. There was no difference of characteristics between the ED and LD group. About 77% (*n* = 72) of analyzed patients were discharged from the hospital within 12 h after RRH (Table 3). Of these, 47 patients (66%) underwent nerve-sparing (NS) RRH (vs. 18% of LD, *p* < 0.001). The ED group had significant correlation with shorter duration for urinary catheter required (1 vs. 39 days, *p* < 0.001), less operative blood loss (100 vs. 125 mL, *p* = 0.004), and less voiding difficulty after long-term follow up (3 vs. 18%, *p* = 0.025) compared to the LD group.

There was no difference of perioperative complications, unexpected visit, and re-admission between the two groups. Most of minor complication included patients who had urinary tract infection requiring antibiotics. The major complication (3.2%, 3/94) included two patients who needed bladder repair and one patient who suffered from vesicovaginal fistula and needed additional surgery. The rate of visit to emergency room after discharge was 21% and 23% in the ED and LD group, respectively. The main reasons for that were voiding difficulty, abdominal pain, and fever. Of these, five patients (7%) of ED group and three patients (14%) of LD group were admitted to hospital again for intravenous antibiotics treatment. We performed univariate and multivariate analyses to identify contributing factors for early discharge of patients after RRH. Performing NS-RRH was only independent predictor for ED (*p* = 0.043, hazard ratio for LD = 0.22, confidence interval = 0.05–0.95).

## 4. Discussion

In this retrospective study, we focused on evaluating quick recovery after RRH under the ERAS setting objectively. Because we had consistent clinical pathway after surgery at our institution, we could evaluate the safety and perioperative outcomes between the ED and LD group in spite of a retrospective design. Our results show that the ED group had feasible perioperative outcomes without the increase of operation-related complications or unexpected visit after surgery compared to the LD group. In addition, NS-RRH was only independent factor for ED after surgery.

Since the ERAS program has been introduced, several studies reported that its implementation in patients who underwent gynecologic cancer surgery was feasible. In 2008, the early period of ERAS, Chase et al. evaluated their ERAS program, including removing urinary catheter at the first day after surgery, early feeding, early ambulation, and prompt conversion to oral analgesics, in 880 gynecologic cancer patients who underwent laparotomy [15]. They reported that ERAS reduced postoperative hospitalization without increasing significant complications. For the application of ERAS protocol in MIS for gynecologic cancers, Chapman et al. evaluated whether ERAS pathway was related to quick recovery and discharge in a retrospective case-control study [16]. The authors analyzed 165 patients undergoing laparoscopic or robotic surgery and showed that ERAS was associated with reduced time for recovery, decreased pain despite reduced opioid use, and lower operative costs.

Although there have been published studies which showed that ERAS program was safe and feasible in gynecologic cancer regardless of the types of surgical approaches, there are few data on quick recovery after surgery in patients with cervical cancer. Because cervical cancer patients who undergo RH, standard treatment for early-stage cervical cancer, have often suffered voiding difficulty after surgery. The incidence of bladder dysfunction following RH has been reported to occur in 24% to 70% [17]. Moreover, long operation time and considerable intraoperative hemorrhage can delay recovery after RH. Therefore, it is difficult to apply same clinical pathways, including quick removal of urinary catheter or early feeding, to all patients who undergo RH. To evaluate enhanced recovery after RH, we needed to focus on the patients who underwent RRH. The technologic advantages of robotic surgery have been expected to help surgeons to perform complex surgical procedures of RH optimally and to reduce operative complications. In addition, some gynecologic surgeons have developed nerve-sparing approach during RH to minimize neurogenic complications [18,19,20]. We also reported the NS-RRH showed early bladder function return and feasible outcomes compared to type C2 RRH in our previous study [10]. Although the present retrospective study could not avoid selection bias completely, it showed NS approach was only independent factor for quick discharge after RRH. We could conclude that minimal bladder function loss after NS-RRH was related to quick recovery after surgery in cervical cancer.

Most of published studies on ERAS program have evaluated the length of hospital stay as one of primary outcomes. Obviously, it is not easy to clarify what are objective values after the implementation of ERAS pathway consisting of early feeding, early ambulation, and multimodal pain control. If we emphasize that ERAS, as the word itself, facilitates quick return to normal activity after surgery, the safety of quick discharge can be evaluated as an objective outcome in the ERAS setting. Based on these considerations, we analyzed perioperative outcomes, postoperative genitourinary function, and the rate of unexpected visit and re-admission to evaluate the feasibility of quick discharge within 12 h after surgery. Our results showed that there was no significant difference between ED and LD cohort. Regardless of the application of ERAS program, there have been several studies on same-day discharge (SDD) after gynecologic cancer surgery. Praiss et al. examined trends and outcomes of SDD for 17,935 endometrial cancer patients who underwent minimally invasive hysterectomy [21]. The authors reported that the rate of readmission did not increase in SDD group. In addition, longer operation time and perioperative complications were related to readmission. For cervical cancer, a Canadian group evaluated the safety of SDD in 119 patients who underwent laparoscopic RH retrospectively [22]. Of these, 63% were SDD patients and they had low risk of postoperative morbidity and hospital readmission. In our study, enrolled patients were categorized as a surgery-to-discharge time of ≤12 h (ED group) or >12 h (LD group) instead of the SDD. Some patients who underwent surgery in the late evening could not go home on that day with or without discomfort. Therefore, we decided that SDD did not reflect all patients who had quick recovery.

Our study has several limitations. Firstly, we could not avoid selection bias completely because this was retrospective study. A well-designed prospective RCT will permit the proper assessment between the ED and LD group. Although several RCTs on ERAS program have been already reported [23,24], it seems to be inappropriate that ERAS program is applied selectively to assigned population because it has already become a part of routine clinical pathways for patients who underwent surgery. Secondly, there is a great debate about whether minimally invasive RH has poor survival outcome in patients with cervical cancer [25,26,27]. However, it is needed to evaluate whether the potential confounding factors, including surgeon’s learning curve or intraoperative tumor spillage can influence on the oncologic outcome after minimally invasive RH [28,29,30]. In addition, a RCT will show the benefits and harms of RRH in cervical cancer separately from laparoscopic RH [31]. Despite these limitations, our study has the strength to support minimal data in describing enhanced recovery after RRH in only cervical cancer patients.

## 5. Conclusions

The ED within 12 h after RRH was safe in the setting of ERAS protocol. The NS-RRH could avoid the delay of genitourinary function recovery after surgery which caused LD. It can become the reasonable clinical pathway to discharge early patients who undergo NS-RRH with ERAS protocol.

## Figures and Tables

**Table 1 jcm-11-01122-t001:** Pain management in the setting of ERAS protocol.

Intraoperative	dexamethasone 12 mg IV
	ketorolac 30 mg IV (do not give with renal insufficiency or elderly patients)
Recovery room	oxycodone/acetaminophen 7.5/325 mg PO
	acetaminophen 1 g IV
	morphine 2 mg IV for breakthrough pain
	ketorolac 30 mg IV if patients staying longer than 6 h
Discharge	oxycodone/acetaminophen 7.5/325 mg PO q 6 h for 2 weeks

IV, intravenous; PO, per oral.

**Table 2 jcm-11-01122-t002:** Clinicopathologic characteristics.

	Early Discharge (*n* = 72)	Late Discharge (*n* = 22)	*p* Value
	Number of Patients (%)		
Age (years)	48.8 ± 13.0	43.1 ± 12.4	0.065
Body mass index (kg/m^2^)	26.5 ± 4.9	28.2 ± 8.2	0.366
Tumor stage			0.477
IB1	61 (84.7)	20 (90.9)	
IB2	11 (15.3)	2 (9.1)	
Tumor size (cm, IQR)	2 (2)	1.3 (3)	0.102
Histology			0.787
Squamous cell carcinoma	39 (54.2)	11 (50)	
Adenocarcinoma	33 (45.8)	10 (45.5)	
Tumor grade			0.938
Well differentiated	22 (30.6)	6 (27.3)	
Moderately differentiated	35 (48.6)	10 (45.5)	
Poorly differentiated	15 (20.8)	6 (27.3)	
Lymphovascular space invasion	20 (27.8)	3 (13.6)	0.167
Parametrium invasion	2 (2.8)	3 (13.6)	0.082
Lymph node metastases	13 (18.1)	3 (13.6)	0.755
Vaginal cuff margin involvement	2 (2.8)	0	1.000

IQR, interquartile range.

**Table 3 jcm-11-01122-t003:** Operative outcomes.

	Early Discharge (*n* = 72)	Late Discharge (*n* = 22)	*p* Value
	Number of Patients (%)		
Nerve-sparing RH	47 (66.2)	4 (18.2)	<0.001
Operating time (min)	189.8 ± 56.1	210.0 ± 48.4	0.132
Estimated blood loss (ml, IQR)	100 (50)	125 (100)	0.004
Number of lymph nodes retrieved	28.0 ± 9.3	26.8 ± 11.2	0.633
Days of urinary catheter required (days, IQR)	1 (21)	39 (37)	<0.001
<1 week	44 (61.1)	4 (18.2)	
1–6 weeks	25 (34.7)	8 (36.4)	
>6 weeks	3 (4.2)	10 (45.5)	
Perioperative complications			
Major	1 (1.4)	2 (9.1)	0.138
Minor	13 (18.1)	6 (27.3)	0.362
Visit to emergency room after discharge	15 (20.8)	5 (22.7)	0.849
Re-admission after discharge	5 (6.9)	3 (13.6)	0.385
Chronic symptoms 2 years after surgery			
Voiding difficulty	2 (2.8)	4 (18.2)	0.025
Overactive bladder	4 (5.6)	2 (9.1)	0.622
Stress urinary incontinence	1 (1.4)	0	1.000

RH, radical hysterectomy; IQR, interquartile range.

## Data Availability

The Excel (Microsoft Corp., Redmond, WA, USA) data used to support the findings of this study were supplied by J.P. under license and requests for access to these data should be made to J.P.

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
