# Peer review of "Enhanced Recovery after Surgery (ERAS) Protocol for Early Discharge within 12 Hours after Robotic Radical Hysterectomy"

_jcm, 2022, doi:10.3390/jcm11041122_

Round 1

Reviewer 1 Report

Abstract

Line 18: NS-RRH miss acronym Line 14-15: miss total number    Introduction: a) Would suggest a brief description of what is ERAS protocol  b) Has been provided sufficient background; the description of the background is good   3. Materials and methods: a) Study design seems not to be mentioned b) Population: overall seems appropriate. Inclusion and exclusion criteria should be described clearly. c) Methods seems adequately described. Criteria for late discharge are classified into minor and major complications. d) Statistics: Appropriately described
e) Missing data is not addressed    Results:
a. Tables and figures are overall easy to understand.  b. the results are clearly presented.   5. Discussion:
a. Clinical relevance: too extensive background description about ERAS protocol, while not in the introduction (see above)
b. Strengths of the study: addressed
c. Limitations of the study: addressed
d. Recommendations for further research: addressed The conclusion are supported by the results.

Author Response

Thank you for your valuable comments and suggestions. We provided responses to your comments below.

Point 1: Line 18: NS-RRH miss acronym.

Response 1: We corrected it properly.

Point 2: Line 14-15: miss total number   

Response 2: We described total number of analyzed patients.

Point 3: Introduction: a) Would suggest a brief description of what is ERAS protocol 

Response 3: We described the following sentences in the ‘Introduction’ as you recommended.

“Enhanced recovery after surgery (ERAS) programs are evidence-based multimodal perioperative protocols designed to get quick recovery after surgery. The integral parts of ERAS protocols consist of preoperative counselling, optimization of nutrition, standardized pain control, and early mobilization [1].”

Point 4: Materials and methods: a) Study design seems not to be mentioned

Response 4: As recommended, we described that we reviewed medical records retrospectively and estimated the patients’ status in the ‘Materials and methods’.

Point 5: Inclusion and exclusion criteria should be described clearly.

Response 5: In this study, we enrolled all cervical cancer patients who underwent robotic radical hysterectomy for study periods. We described the following sentence in the ‘Materials and methods’ to clarify the inclusion and exclusion criteria as you recommended.

“Between April 2008 and July 2018, we performed RRH in 94 patients with cervical cancer. In this study, all of them were enrolled without exception…”

Point 6: Discussion: a. Clinical relevance: too extensive background description about ERAS protocol, while not in the introduction (see above)

Response 6: As recommended, we added the description of what was ERAS protocol in the ‘Introduction’ briefly. In addition, we omitted the redundant descriptions about results from ERAS study in the ‘Discussion’.

Reviewer 2 Report

The paper is devoted to the problem of early discharge after radical hysterectomy. Additional circumstaces are applied robotic surgery, ERAS protocol , and nerve sparing technique[in some patients]. The very interesting and important question is how the authors decided who may leave the hospital before 12 hours. The authors only mention that they had  "consistent clinical pathway after surgery at their institution" but did not supply with its details.   The easy recognized defect of the paper is lack of Table 4.

Author Response

Thank you for your valuable comments and suggestions. We provided responses to your comments below.

Point 1: The paper is devoted to the problem of early discharge after radical hysterectomy. Additional circumstances are applied robotic surgery, ERAS protocol, and nerve sparing technique [in some patients]. The very interesting and important question is how the authors decided who may leave the hospital before 12 hours. The authors only mention that they had "consistent clinical pathway after surgery at their institution" but did not supply with its details.  

Response 1: According to the perioperative care protocol in our institution, the urinary catheters are removed immediately after surgery in all patients who undergo robotic radical hysterectomy. After that, we decide patients to leave the hospital according to their voiding pattern. In this study, the time of 12 hours was just a point to divide study population into two cohorts, including early and late discharge. We described the relevant sentences with details in the ‘Materials and methods’.

Point 2: The easy recognized defect of the paper is lack of Table 4.

Response 2: In the Table 4, both the continuous and categorical factors were evaluated in the univariate and multivariate analysis. Although the continuous factors were not categorized for multivariate analysis, we regarded them as integral variables to affect the time for discharge and included their results in the Table 4.